# Polyacrylamide Regulated Phytohormone Balance and Starch Degradation to Promote Seed-Potato Sprouting and Emergence

**DOI:** 10.3390/plants13192796

**Published:** 2024-10-05

**Authors:** Meiqiang Yin, Dongmei Hu, Xiaohang Yu, Yijie Wang, Sa Song, Chongyue Wang, Qilin Hu, Yinyuan Wen

**Affiliations:** College of Agronomy, Shanxi Agricultural University, Taigu 030801, China; yinmq999@163.com (M.Y.); hdm16634254580@126.com (D.H.); 17836507791@163.com (X.Y.); 13038000386@163.com (Y.W.); songsa0103@163.com (S.S.); 18387097921@163.com (C.W.); 18110600776@163.com (Q.H.)

**Keywords:** polyacrylamide, potato, sprouting, phytohormone, starch conversion

## Abstract

Potatoes are typically seeded as tubers, and their slow sprouting significantly impacts production. Therefore, the effects of polyacrylamide (20 g·L^−1^, 30 g·L^−1^, and 40 g·L^−1^) as a seed potato dressing on sprouting, seedling growth, and biomass were investigated. The phytohormone content, respiratory intensity, and starch metabolism enzyme activity were analyzed to elucidate the physiological mechanisms involved. The sprouting rate significantly increased after 20 g·L^−1^ and 30 g·L^−1^ treatments by 40.63% and 15.63%, respectively. The sprouting energy was the highest (52.0%) at 20 g·L^−1^, 7.67 times higher than the control. The 20 g·L^−1^ and 30 g·L^−1^ treatments also promoted emergence and growth, with the emergence rate increasing by 18.18% and 27.27% and growth increasing by over 8.1% and 11.9%, respectively. These effects were related to changes in phytohormone content and accelerated starch conversion. After treatment, the auxin and cytokinin contents in the apical buds increased significantly at the germination initiation stage, and during the germination and vigorous growth phases, the auxin, cytokinin, and gibberellin contents increased. Polyacrylamide treatment activated α-amylase and promoted starch degradation, increasing soluble sugar content to provide nutrients and energy for sprouting. This study provides a promising approach for promoting potato tuber sprouting and seedling growth.

## 1. Introduction

Potatoes (*Solanum tuberosum* L.) are a staple food for two-thirds of the world’s population and are one of the top four crops worldwide [1]. Potatoes are highly adaptable crops that can grow and be harvested under adverse conditions such as poverty and drought [2]. Therefore, they play crucial roles in global food and human nutritional security. The greatest obstacle to the widespread development of potatoes as a bulk crop is the difficulty of controlling seed tuber sprouting. Depending on the intended purpose, accelerated (i.e., seed tuber) or delayed (i.e., industrial processing) sprouting of harvested tubers is favorable. Immediately after harvest, potato tubers cannot be induced to sprout because of their dormancy, defined as the physiological state in which autonomous sprout growth does not occur even when the tuber is placed under ideal conditions for sprout growth. Potato tuber dormancy must be broken before the tubers can be used as seed potatoes. Dormancy release in potato tubers is closely determined by both genetic and environmental factors such as soil water, temperature, light, and soil texture [1]. The sprouting speed and ability of seed potatoes are closely related to the uniformity and strength of seedlings and play a key role in maintaining yield. In potato production practices, problems such as delayed emergence, non-uniform emergence, and low emergence rates often occur, greatly affecting yield. Hence, it is essential to develop methods that promote potato sprouting to achieve rapid, uniform, and strong seedling emergence [3].

Multiple measures have been developed to promote potato sprouting, including physical and chemical approaches. Thermal shock boosts enzyme activity in tubers, stimulates faster development of sprouts from the eyes, reduces the sprouting period, and accelerates plant development and tuber formation [4]. Exposure to blue light (460 nm) can shorten the dormancy period of microtubers, whereas red light (660 nm) and far-red light (735 nm) are the most effective in suppressing sprout growth [5,6]. Magnetic fields, cold plasma, and ultraviolet (UV)-C have been used to regulate potato sprouting and dormancy [7]. Complex interactions between hormones play a crucial role in regulating potato dormancy-to-sprouting transition processes [8]. Gibberellins (GA) and cytokinins (CTK) are generally considered growth promoters, whereas abscisic acid (ABA) and ethylene are believed to inhibit sprout growth [1]. Exogenous application of GA accelerates seed potato sprouting and ensures vegetative growth by facilitating starch breakdown and sucrose synthesis [9,10]. Overexpression of GA20-oxidase in transgenic potatoes resulted in accelerated sprout growth [11]. Cytokinins are involved in the early stages of sprouting and induce the termination of dormancy, whereas GA promotes sprout growth [12]. The interaction between CTK and GA underscores the delicate hormonal equilibrium necessary for sprouting initiation [7]. The ABA content in tubers is usually high at the onset of dormancy, peaks during deep dormancy, and declines during post-harvest storage, with a significant rise in auxin (IAA) levels, which coincides with the break of dormancy [1,13]. The transition from dormancy to sprouting is marked by the degradation of ABA, tipping the balance toward the actions of other growth-promoting phytohormones such as GA and CTK [7,14]. Auxin stimulates bud development by hastening cell growth and differentiation processes that underlie dormancy breaking and sprouting [15]. Sprout promotion can be achieved commercially through the application of chemical compounds. Carbon disulfide (CS2) accelerates tuber sprouting and increases tuber weight loss [16]. Amphiphilic nanocomposites can endow potatoes with an appropriate occupational area water-to-air ratio, effectively promoting seed potato sprouting and increasing yield [3].

Polyacrylamide (PAM) possesses unique characteristics that include high water solubility, high viscosity, and effective flocculation capabilities. These properties make it valuable in various sectors, such as agriculture, wastewater treatment, enhanced oil recovery, and mineral processing [17]. Polyacrylamide is a strongly hydrophilic gel-forming compound that absorbs up to 2000 times its weight in water [18,19]. Polyacrylamide enhances soil water-holding capacity and improves soil physical properties, thus increasing microbial biomass carbon, nitrogen mass fraction, and crop yield [20]. Soaking foxtail millet (*Setaria italica* L.) seeds in an appropriate concentration of PAM effectively promoted seed germination and seedling growth under drought stress [21]. Oxidized sodium alginate/polyacrylamide hydrogels significantly improve survival time and promote wheat seedling growth in brown loam soil in water-deficient environments [22]. The application of water-retaining agents can increase soil saturation and reduce soil particle density and bulk density, thereby increasing crop productivity [23], and can also effectively promote seed potato germination and growth [3]. These results indicate that PAM can serve as a soil water-retaining and seed-priming agent, promoting germination and seedling growth, enhancing stress resistance, and increasing yield.

In this study, different concentrations of PAM were used as seed potato dressings, and their effects on sprouting, seedling emergence and growth, and biomass were investigated. Subsequently, the phytohormone content, respiratory intensity, and enzyme activity of starch metabolism were analyzed to elucidate the mechanisms causing the observed effects. This work provides a simple, low-cost approach for enhancing seed potato sprouting, which may have broad potential applications in potato production.

## 2. Results

### 2.1. PAM Treatment Promoted Potato Sprouting in Air

Polyacrylamide significantly increased potato sprouting rate and speed (Figure 1 and Figure 2). On the 15th day, potato sprouting rates under the PAM_20_, PAM_30_, and PAM_40_ treatments were 40%, 30%, and 20%, respectively, while CK potatoes had not yet sprouted (Figure 1). Compared to CK, on the 27th day, the final sprouting rates under PAM_20_ and PAM_30_ treatments increased significantly by 40.63% and 15.63%, respectively, whereas the PAM_40_ treatment sprouting rate was lower than that of the CK.

Sprouting energy is a component of seed vigor that corresponds to the speed and uniformity of seedling emergence. Potato seed sprouting energies were increased significantly by PAM treatment to a maximum of 52.00% with PAM_20_, 7.67 times higher than CK (Figure 3).

The PAM_20_ and PAM_30_ treatments both enhanced sprout quality, as demonstrated by an increase in bud length and diameter; however, there was no significant difference between the two treatments (Figure 4). For example, on the 39th day, the PAM_20_ and PAM_30_ treatment bud lengths were 10.24 mm and 10.41 mm, with increases of 48.19% and 50.86%, respectively, over CK (Figure 4A). Sprout diameter was the highest in the PAM_30_ treatment during the entire sprouting stage (Figure 4B). On the 70th day, the sprout diameter in the PAM_30_ treatment was 3.14 mm, 17.43% higher than CK.

### 2.2. PAM Treatment Increased Potato Seedling Emergence

The PAM treatment promoted potato seedling emergence (Figure 5 and Figure 6). After treatment with PAM_20_ and PAM_30_, potato seedling emergence rates were 86.67% and 93.33%, respectively, significantly higher than CK by 18.18% and 27.27%, respectively. Seedling plant height, stem diameter, and fresh/dry weight of potatoes subjected to the different treatments were in the following order: PAM_30_ > PAM_20_ > PAM_40_ > CK (Table 1, Figure 7). The effects of PAM_20_ and PAM_30_ were very significant: plant height increased by 8.1% and 15.3%, stem diameter increased by 11.9% and 20.0%, respectively, and biomass (including shoot and root fresh/dry weight) increased by >10.5%.

### 2.3. PAM Treatment Enhanced Potato Tuber Respiration Intensity

During the sprouting phase, potato tuber respiration intensity increased from the 7th to the 21st day (Figure 8). All three PAM treatments significantly enhanced potato tuber respiration intensity on the 7th day. At the 14th and the 21st days, PAM_20_ and PAM_30_ significantly increased sprouting potato tuber respiration, whereas the PAM_40_ treatment did not significantly differ from CK. The respiration intensity in potatoes treated with PAM_20_ was the highest on the 14th day at 23.55 mg·kg^−1^·h^−1^, a 69.61% increase compared to CK.

### 2.4. PAM Treatment Altered Endogenous Phytohormone Levels in Potato Tuber Apical Buds

During the sprouting phase, the IAA (Figure 9A), GA (Figure 9B), and CTK (Figure 9C) contents in potato tuber apical buds gradually increased, whereas the ABA (Figure 9D) content showed no significant change. Polyacrylamide treatment altered these endogenous phytohormone levels. During the GI phase, PAM treatment had no effect on GA and ABA content. Auxin and CTK contents increased significantly with PAM_20_ and PAM_30_ at the GI stage, and the maximum contents obtained with PAM_30_ treatment were increased by 12.94% and 47.04%, respectively, compared to CK.

During the GS and VG phases, the phytohormone (IAA, GA, and CTK) content in potato tuber apical buds treated with PAM_20_ and PAM_30_ was significantly higher than that in CK and PAM_40_. More specifically, the GA content in the PAM_20_ and PAM_30_ treatments was significantly higher than that in other treatments, with increases of 19.19% (GS phase) and 22.32% (VG phases), respectively, when compared to CK. The CTK levels in the PAM_20_ and PAM_30_ treatments were significantly higher than those in other treatments, with the PAM_30_ treatment showing the highest levels during the GS phase, an increase of 47.04% over CK. The IAA content in the PAM_30_ treatment was significantly elevated compared to other treatments during both the GS and VG phases, showing increases of 18.33% and 23.95%, respectively, compared to CK.

During the GS and VG phases, PAM treatment reduced the ABA content. At the GS stage, the effect of PAM treatment on ABA levels was not significant. At the VG stage, ABA content after PAM_20_, PAM_30_, and PAM_40_ treatments was significantly lower than that of CK, decreasing by 35.43%, 34.84%, and 35.66%, respectively.

### 2.5. PAM Treatment Accelerated Starch Transformation by Activating α-Amylase

When seed potato tubers entered the sprouting phase, the α-amylase activity increased sharply (Figure 10A), and the starch content decreased steadily (Figure 10B). All three PAM concentrations stimulated α-amylase activity. On the 7th day, α-amylase activity was highest after PAM_20_ treatment, whereas PAM_40_ treatment maximized α-amylase activity on the 21st day, increasing it by 65.30%. By the 14th day, the starch content in the PAM_30_ and PAM_40_ treatments was reduced by 12.2% and 32.1%, respectively, compared to CK.

During the sprouting phase, starch is transformed into soluble sugar (including reducing sugar) and continuously increased to provide nutrients and energy for potato sprouting (Figure 10C,D). The soluble sugar and reducing sugar contents in potato tubers were increased by PAM treatment (Figure 10C,D). By the 14th day, PAM_30_ maximized reducing sugar content, increasing it by 23.4%. However, on the 21st day, the reducing sugar content in the PAM_20_ treatment peaked, showing an increase of 30.92% compared with CK (Figure 10C). The soluble sugar content significantly increased after PAM treatment, and the effect of PAM_20_ was the most significant, increasing by 15.98%, 20.90%, and 16.43% on the 7th, 14th, and 21st days, respectively.

## 3. Discussion

### 3.1. PAM Treatment Promoted Potato Sprouting and Seedling Growth

When potato tubers are used as seeds, sprouting must be accelerated to achieve rapid, uniform, and strong seedling emergence [1,3]. Several methods have been used to promote seed potato germination. Potato tubers immersed in 1 μm/L γ-aminobutyric acid promoted tuber sprouting and POD42 protein levels [24]. Applying GA or Carbon disulfide (CS_2_) advanced the breaking of dormancy associated with the removal of apical dominance and an increase in the number and dry matter content of sprouts [25]. Brassinosteroids significantly disrupt dormancy and enhance the sprouting ability of potato tubers [26]. Nano-priming is a considerably more effective technology that helps to improve seed germination, seedling growth, and yield in agriculture [27]. Attapulgite/H-SiO_2_ nanocomposites can efficiently increase the number and length of potato sprouts and promote the growth of potato seedlings by increasing plant height, seedling number, and stem diameter [3]. Polyacrylamide, a potent hydrophilic gel-forming synthetic polyelectrolyte containing hydrophilic groups, is used in many industries, including cosmetics, pharmaceuticals, food, agriculture, and wastewater treatment [17]. Foxtail millet seeds soaked in PAM exhibited increased germination energy, germination rate, seedling height, and root length under drought stress [21]. Oxidized sodium alginate/polyacrylamide hydrogels lock large amounts of water; thus, wheat plants are able to absorb water and grow in brown loam soil in water-deficient environments [22]. In the present study, the PAM_20_ and PAM_30_ treatments improved seed potato tuber vigor (sprouting energy) and increased their sprouting rate and speed (Figure 3). The PAM treatment also enhanced sprout quality and increased potato bud length and diameter (Figure 4). The potato seedling emergence rate, seedling plant height, stem diameter, and fresh/dry weight were all increased by PAM (Figure 5 and Figure 6; Table 1). These results indicate that PAM can serve as a seed potato tuber-priming agent to accelerate potato emergence under field conditions. Additionally, PAM is likely to be categorized as a low-risk chemical when employed as a seed priming agent or soil amendment, while further experiments are crucial for the potential benefit and risk assessments [17].

### 3.2. PAM Regulated Phytohormones during Potato Tuber Sprouting

The duration of potato tuber dormancy depends on the genetic and environmental factors. Termination of dormancy and promotion of sprouting are believed to be primarily under hormonal control [28]. Abscisic acid is associated with the onset and maintenance of tuber dormancy, whereas GA and CK are thought to be involved in the release of dormancy [29]. At harvest, the ABA content in tubers is usually high and then gradually decreases until dormancy is released. Dormancy release was accelerated in StCYP707A-overexpressing potato lines because of their lower ABA and higher GA levels. The ABA content of different potato varieties showed a continuous decline during storage, but the absolute ABA levels did not affect the potato sprouting [29]. Polyacrylamide had no effect on ABA content at the GI and GS stages but significantly decreased it in the VG phase (Figure 9), indicating that potato tuber dormancy release and spouting may not be controlled by reducing ABA levels [28,30]. Gibberellins act as ABA antagonists in regulating bud dormancy and sprouting. The exogenous application of bioactive GA (GA_1_ and GA_20_) accelerated the sprouting of seed potatoes and ensured vegetative growth, whereas GA_12_, GA_19_, and GA_8_ were inactive during dormancy release. Higher levels of GA significantly promote vigorous sprout growth. In this study, after PAM treatment, the GA content did not increase at the GI stage but significantly increased in the GS and VG phases. It was concluded that PAM treatment restored endogenous GA content to promote sprout growth after the termination of dormancy (GS and VG phases) but did not break tuber dormancy (GI) [29]. This is because the reactivation of meristem activity (or dormancy release) and sprout growth in potato tubers requires both GA and CTK [29]. Potato tubers with decreased CTK levels do not respond to GA application; in contrast, tubers with increased CTK content promote GA-induced sprouting [31]. The application of exogenous CTK shortens the dormancy phase and triggers earlier sprouting, which is consistent with an increase in endogenous CTK (isopentenyl adenine and trans-zeatin) concentrations [32]. Polyacrylamide treatment increased CTK levels in potato tubers and apical buds, especially in the GI phase, which promoted sprout onset and growth together with GA in the GS and VG phases (Figure 9). Auxin enhanced dormancy breaking in potato tubers through a gradual increase in concentration in the bud eyes, with a significant increase in free IAA observed from the point of harvest to the end of dormancy. The breaking dormancy of potato tubers coincides with a significant rise in free IAA in the eyes [33]. Auxins play an important role in the onset of tuber sprouting by stimulating vascular tissue differentiation and determining leaf primordia formation in newly developing sprouts [30]. Additionally, the expression of auxin response factor 6 (ARF6) is strongly upregulated during dormancy release, especially in the developing vasculature, and ARF6 has been identified as a marker of meristem reactivation in potato tubers [34]. In the present study, PAM treatment increased the IAA content in potato tubers and apical buds at the GI stage and, together with CTK, initiated bud sprouting (Figure 9). Taken together, these results suggest that these hormones may serve as endogenous signals for tubers to terminate dormancy and initiate growth [30]. That is to say, PAM_20_ and PAM_30_ treatment induced increases in IAA and CTK to promote the shift from potato tuber dormancy to activation and sprout initiation; following the GS and VG phases, bud sprouting and elongation resulted from the interactions between the rise of promoting phytohormone groups (GA, IAA, and CTK) and the diminution of ABA. However, the mechanisms underlying phytohormone coordination during tube sprouting after PAM treatment require further exploration. However, the molecular mechanisms of hormones and their interactions in PAM-promoting potato sprouting should be further investigated to obtain a more complete pathway.

### 3.3. PAM Accelerated Starch Degradation to Provide Nutrients for Potato Tuber Sprouting

It is generally accepted that tuber sprouting is accompanied by some metabolic changes, including the degradation of starch and protein, increases in soluble sugars and amino acids, and variations of endogenous phytohormones, as well as some alterations in relevant gene transcripts and enzymatic reactions [34]. Once dormancy breaks, the sprouts begin to grow. These sprouts rely on the mother tuber for energy, nutrients, and other substances required for growth. Simultaneous silencing of isoamylases ISA1, ISA2, and ISA3 in potato tubers leads to a reduction in starch content and changes in glucan and granule structure, which makes the granules more soluble and accessible to degrading enzymes, leading to the production of sufficient amounts of sucrose during storage to initiate early sprouting [35]. The expression of bacterial pyrophosphatase accelerates potato tuber sprouting due to the enhanced conversion of starch to sucrose and/or other cell wall precursors [36]. The α-amylase StAmy23 was involved in starch breakdown and stimulated sprouting by hydrolyzing soluble phytoglycogen to ensure the supply of sugars [37]. Glucose, sucrose, and fructose contents are positively correlated with the termination of dormancy [38]. Gamma-aminobutyric acid promotes sprouting by increasing soluble sugar content and decreasing starch content [24]. In this study, PAM treatment stimulated the α-amylase activity, hastened starch degradation, and accelerated the production of soluble sugars (including reducing sugar) to provide nutrients for potato sprouting (Figure 10).

### 3.4. PAM Enhanced Respiration to Provide Energy for Potato Tuber Sprouting

Respiration converts nutrient reserves to form energy and fuel cell division and the production of new cells. The respiration rate of tubers was positively correlated with sprouting and sprout growth of tubers during storage, and respiration appears to be the pacemaker of physiological age [39]. Transgenic potato plants expressing an additional inorganic pyrophosphatase gene sprout earlier because they have a much-increased general metabolic activity characterized by a massive increase in glycolysis and respiration [40,41]. Potato tuber dormancy breakage by CS_2_ (Carbon disulfide) treatment was associated with an increase in the respiration rate and the contents of sucrose, fructose, and glucose [16]. Electron beam irradiation prolongs potato tubers’ dormancy period by inhibiting respiration and regulating the metabolism of membrane lipid peroxidation and antioxidants [42]. Potato sprout suppressor chlorpropham (CIPC) inhibits sprouting by reducing respiration and carbohydrate metabolic activity [28]. Amphiphilic nanocomposites provide sufficient air but insufficient water onto the potato surface, boosting respiration and resulting in worse sprouting [3]. PAM treatment enhanced potato tuber respiration intensity with dose dependency, and PAM_20_ and PAM_30_ significantly increased sprouting potato tuber respiration, whereas PAM40 did not significantly affect it (Figure 8). The results indicated that PAM treatment enhanced potato tuber respiration intensity and carbohydrate conversion to provide energy and nutrients for potato tuber sprouting. While PAM treatment enhanced potato respiration by cytochrome oxidase pathway or alternative oxidase pathway should be further investigated to determine a clear mechanism.

## 4. Materials and Methods

### 4.1. Materials

Jinshu 16 is a medium-late maturing potato variety. Its dry matter and starch content were 22.3% and 15.6%, respectively, which were used as processing potatoes and fresh vegetable potatoes. Disease-free potato mini-tubers (15–25 g each potato) were purchased from Kangnong Shuye Co., Ltd. (Lvliang, Shanxi, China). Polyacrylamide was purchased from Jingfen Huagong Co., Ltd. (Shenyang, China).

### 4.2. Seed Potato Treatment with PAM

Two hundred milligrams of potato mini-tubers were mixed with 300 mL PAM solution (0, 20, 30, 40 mg·L^−1^; CK, PAM_20_, PAM_30_, and PAM_40_, respectively), soaked, and stirred for 5 min. The mixture of potatoes and PAM (dressed tubers) were then transferred to the dark room for air-drying and used for sprouting observations in air and emergence investigations in pots.

### 4.3. Potato Sprouting in Air after PAM Treatment

Twenty air-dried, treated seed potatoes were transferred to a dark culture room (70% humidity, 25 °C) for sprouting observations. Potatoes with sprouts of >2 mm were considered sprouted. The sprouted potato numbers were counted daily to calculate the sprouting rate, and the sprouting energy was counted on the 17th day. The longest sprout length and diameter were measured every 7 days from the 18th to 70th days. The respiratory intensity (mg CO_2_·kg^−1^·h^−1^) was measured using a photosynthesis system (TP-3051D; Zhejiang Top Yunnong Technology Co., Ltd., Hangzhou, China) on the 7th, 14th, and 21st days, along with α-amylase activity, starch, reducing sugar, soluble sugar, and sucrose content. All tests were performed in triplicate.

### 4.4. Potato Sprouting in Soil after PAM Treatment

Thirty air-dried, treated seed potatoes were planted in rectangular (50 × 19 × 14 cm) pots in 1.2 cm^3^ of soil (humidity of 70%) at a depth of 10 cm. Five potatoes were sown per pot, with six pots per PAM treatment. The pots were then placed in a greenhouse (14 h light and 10 h dark, relative humidity of 60%) at 25 °C, and the soil moisture level was maintained at 65% by daily weighing and water replenishment. The emergence rate was calculated on the 15th day after sowing. Plant height, stem diameter, and shoot biomass were determined on the 40th day. All tests were performed in triplicate.

### 4.5. Quantitative Analysis of Phytohormones

Phytohormone levels were analyzed at four distinct stages of potato sprouting. Before treatment (BT), at germination initiation (GI), during germination (GS), and during vigorous growth (VG), which were 0, 20, 30, 40 days after PAM treatment, respectively. The meristematic tissue (0.5 g) of the top bud and tuber bud eye tissue were frozen with liquid nitrogen and ground in 4.5 mL 0.01 mol · L^−1^ phosphate buffer solution (pH 7.5). Then, the mixture was centrifuged (4 °C) at 12,000× *g* for 10 min. The supernatant was used to quantify endogenous phytohormone levels using an enzyme-linked immunosorbent assay (Shanghai Enzyme Union Biotechnology Co., Ltd., Shanghai, China).

### 4.6. Starch Metabolism Analysis

Sprouting tuber tissues were used to analyze starch metabolism. The 3,5-dinitrosalicylic acid colorimetric method was used to determine the activity of α-amylase and the reducing sugar content. Starch and soluble sugar contents were quantified using the anthrone-sulfuric acid colorimetric method.

Dried tuber tissues were ground and passed through a 0.15 mm sieve. The powdered sample (0.2 g) was placed in 6 mL 80% (*v*/*v*) ethanol and then incubated in 80 °C water bath for 30 min. Then, the mixture was centrifuged (4 °C) at 4000× *g* for 5 min, the supernatant was collected, 6 mL of 80% ethanol was added to the residue, and the extraction was repeated twice. The three supernatants were collected and diluted to 25 mL using 80% ethanol. The resulting extract was used to determine the soluble and reducing sugar content by spectrophotometry at OD_620_ and OD_540_.

The ethanol-insoluble residue was then stored for starch extraction. The starch in the residue was released by boiling with 2 mL of distilled water for 15 min and cooling to 10 °C. Then, the starch was hydrolyzed with 9.2 mol·L^−1^ HClO_4_ (2 mL) for 15 min. Four milliliters of distilled water was added to the samples, which were centrifuged at 4000× *g* and 4 °C for 10 min. The residue was extracted again using 4.6 mol·L^−1^ HClO_4_ (2 mL). The supernatants were combined and diluted with distilled water to 25 mL to obtain starch extract. The starch content was measured using a spectrophotometer at OD_620_.

Potato tuber tissues (1.0 g) ground using a chilled mortar containing 1 mL of pre-cooled distilled water. The slurry was centrifuged at 3000× *g* at 4 °C for 10 min. One milliliter of supernatant was used to assay the α-amylase activity by 3,5-dinitrosalicylic acid colorimetric spectrophotometry at OD_540_.

### 4.7. Statistical Analysis

All figures were drawn using the Origin 2022 software (https://www.originlab.com/2022, accessed on 18 March 2024). SPSS 25.0 (https://www.ibm.com/spss, accessed on 18 May 2024) was used to analyze the differences among the independent sample *t*-test, analysis of variance (ANOVA), and multiple comparisons (Duncan) (*p* = 0.05).

## 5. Conclusions

Polyacrylamide treatment of potato tubers increases the content of growth-promoting phytohormones (IAA, CTK, and GA), reduces ABA content to break dormancy, and initiates sprouting. Amylase is activated by PAM to promote starch degradation and its conversion into soluble and reducing sugars. Furthermore, PAM enhances the respiration rate of potato tubers during the sprouting phase. These physiological changes provide nutrients and energy for the growth of sprouting potato seedlings. The optimal treatments were found to be PAM_20_ and PAM_30_. This study provides a promising approach for promoting potato tuber sprouting and seedling growth with broad practical application in promoting potato tuber sprouting. The potential challenges to industrial-scale application of PAM-promoting sprouting in potatoes were carefully considered, including (i) differences in response among varieties and (ii) the compatibility of PAM with seed coating agents and plant regulators.

## Figures and Tables

**Figure 1 plants-13-02796-f001:**
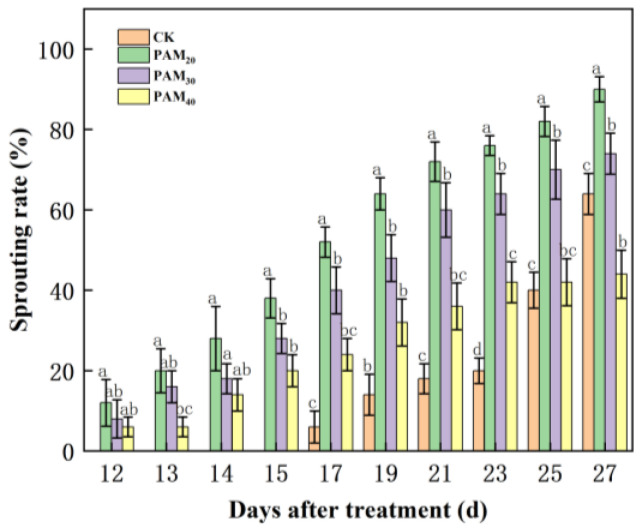
Effect of polyacrylamide (PAM) treatment on sprouting rates of potato mini-tubers. Note: CK, control (no polyacrylamide treatment); PAM_20_, 20 mg·L^−1^ polyacrylamide; PAM_30_, 30 mg·L^−1^ polyacrylamide; PAM_40_, 40 mg·L^−1^ polyacrylamide. Lowercase letter above the diagram columns indicates a significant difference (*p* < 0.05) among treatments. The data represent the mean ± SD of three biological replicates.

**Figure 2 plants-13-02796-f002:**
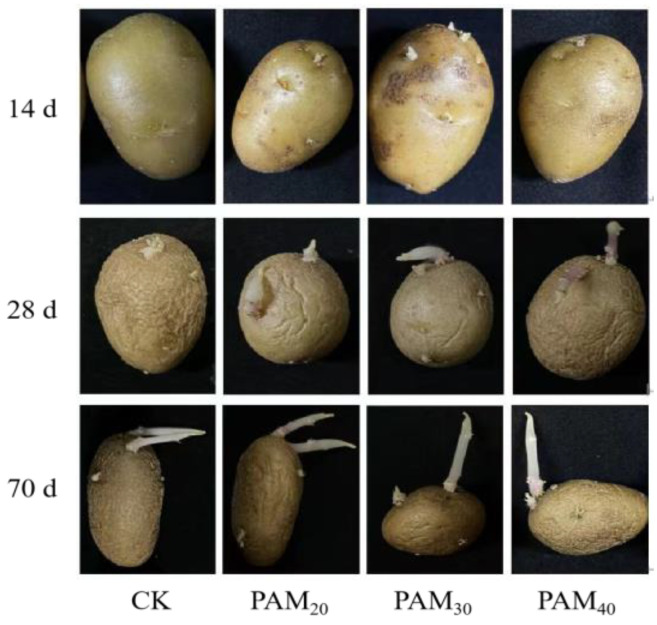
Potato mini-tuber sprouting and bud growth performance treated by PAM treatment. Note: CK, control (no polyacrylamide treatment); PAM_20_, 20 mg·L^−1^ polyacrylamide; PAM_30_, 30 mg·L^−1^ polyacrylamide; PAM_40_, 40 mg·L^−1^ polyacrylamide.

**Figure 3 plants-13-02796-f003:**
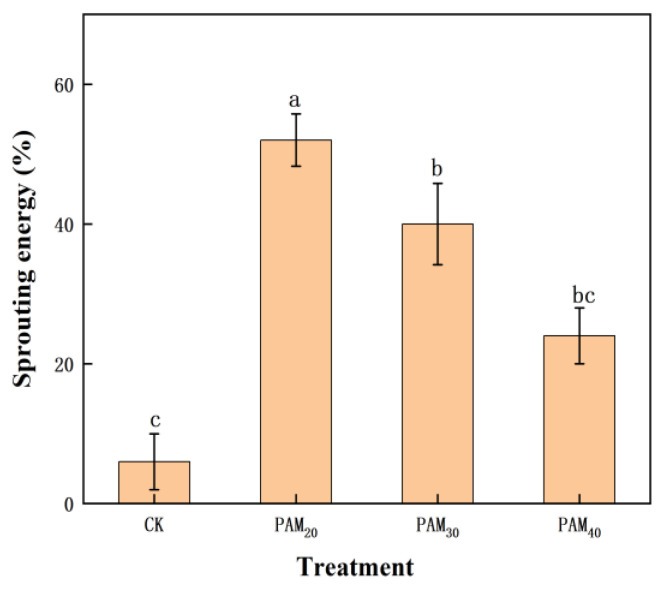
Effect of PAM treatment on sprouting energy of potato mini-tubers. Note: CK, control (no polyacrylamide treatment); PAM_20_, 20 mg·L^−1^ polyacrylamide; PAM_30_, 30 mg·L^−1^ polyacrylamide; PAM_40_, 40 mg·L^−1^ polyacrylamide. Lowercase letter above the diagram columns indicates a significant difference (*p* < 0.05) among treatments. The data represent the mean ± SD of three biological replicates.

**Figure 4 plants-13-02796-f004:**
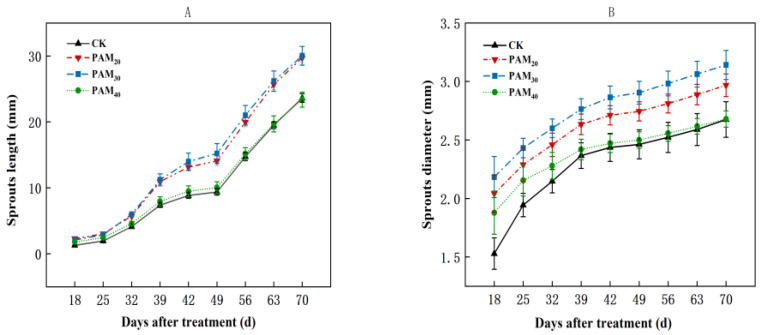
Effect of PAM treatment on the sprout length and diameter of potato sprouting in air. (**A**) Sprout length. (**B**) Sprout diameter. Note: CK, control (no polyacrylamide treatment); PAM20, 20 mg L^−1^ polyacrylamide; PAM30, 30 mg L^−1^ polyacrylamide; PAM40, 40 mg L^−1^ polyacrylamide. The data represent the mean ± SD of three biological replicates.

**Figure 5 plants-13-02796-f005:**
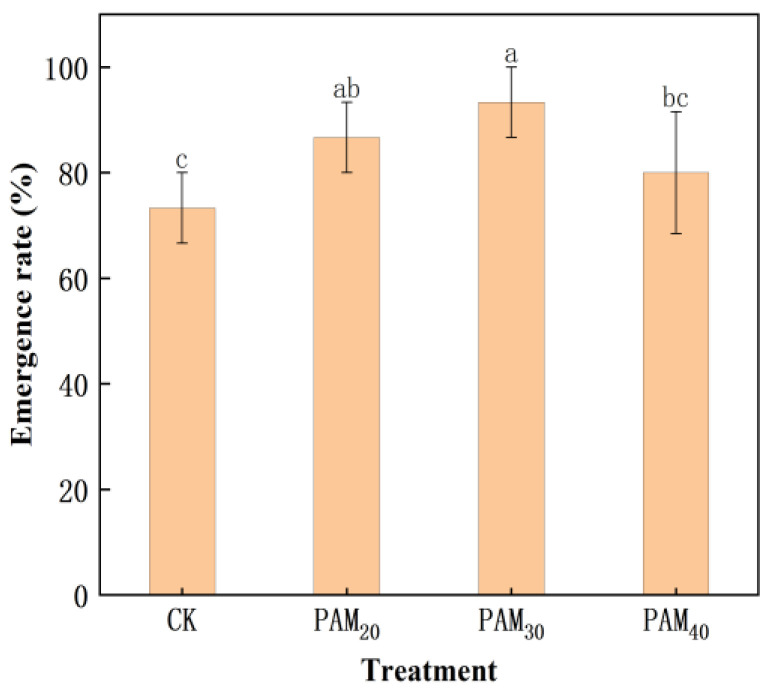
Emergence rate of potted potato mini-tubers treated with PAM. Note: CK, control (no polyacrylamide treatment); PAM_20_, 20 mg·L^−1^ polyacrylamide; PAM_30_, 30 mg·L^−1^ polyacrylamide; PAM_40_, 40 mg·L^−1^ polyacrylamide. Lowercase letter above the diagram columns indicates a significant difference (*p* < 0.05) among treatments. The data represent the mean ± SD of three biological replicates.

**Figure 6 plants-13-02796-f006:**
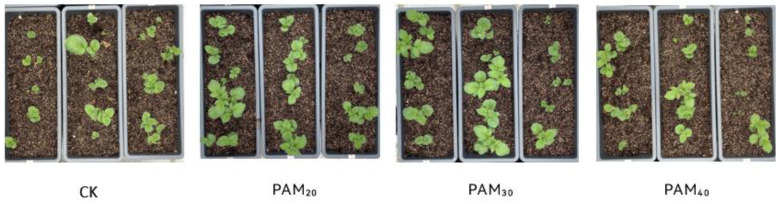
Effect of PAM on the emergence of potted potato mini-tubers. Note: CK, control (no polyacrylamide treatment); PAM_20_, 20 mg·L^−1^ polyacrylamide; PAM_30_, 30 mg·L^−1^ polyacrylamide; PAM_40_, 40 mg·L^−1^ polyacrylamide.

**Figure 7 plants-13-02796-f007:**
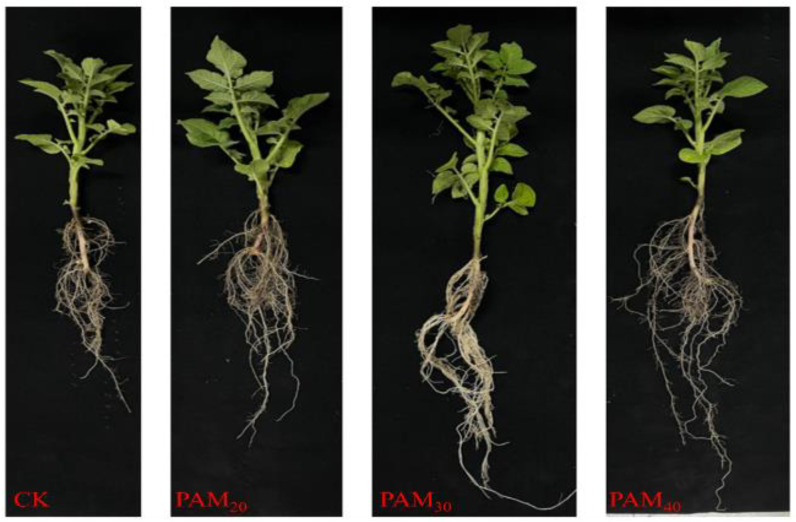
Effect of PAM on seedling growth of potted potato mini-tubers. Note: CK, control (no polyacrylamide treatment); PAM_20_, 20 mg·L^−1^ polyacrylamide; PAM_30_, 30 mg·L^−1^ polyacrylamide; PAM_40_, 40 mg·L^−1^ polyacrylamide.

**Figure 8 plants-13-02796-f008:**
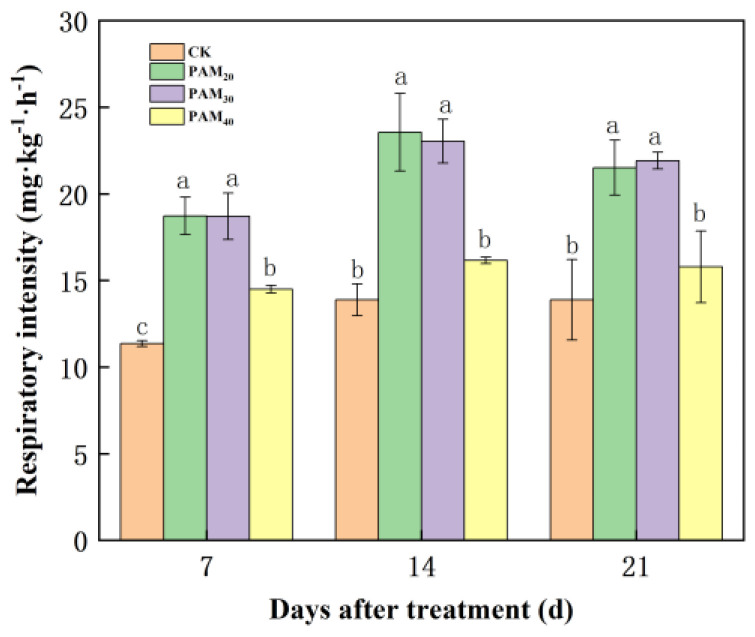
Effect of PAM on the respiratory intensity of potato mini-tubers during the sprouting phase. Note: CK, control (no polyacrylamide treatment); PAM_20_, 20 mg·L^−1^ polyacrylamide; PAM_30_, 30 mg·L^−1^ polyacrylamide; PAM_40_, 40 mg·L^−1^ polyacrylamide. Lowercase letter above the diagram columns indicates a significant difference (*p* < 0.05) among treatments. The data represent the mean ± SD of three biological replicates.

**Figure 9 plants-13-02796-f009:**
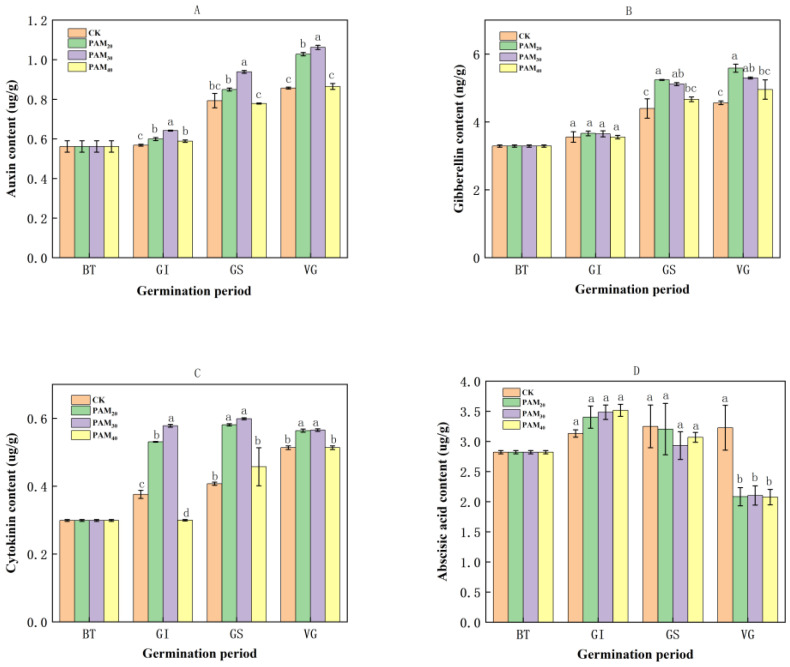
Effect of PAM treatment on endogenous phytohormone content in potato tuber terminal buds. (**A**) Auxin content. (**B**) Gibberellin content. (**C**) Cytokinin content. (**D**) Abscisic acid content. Note: BT: Before treatment; GI: Germination initiation; GS: Germination stage; VG: Vigorous growth. CK, control (no polyacrylamide treatment); PAM_20_, 20 mg·L^−1^ polyacrylamide; PAM_30_, 30 mg·L^−1^ polyacrylamide; PAM_40_, 40 mg·L^−1^ polyacrylamide. Lowercase letter above the diagram columns indicates a significant difference (*p* < 0.05) among treatments. The data represent the mean ± SD of three biological replicates.

**Figure 10 plants-13-02796-f010:**
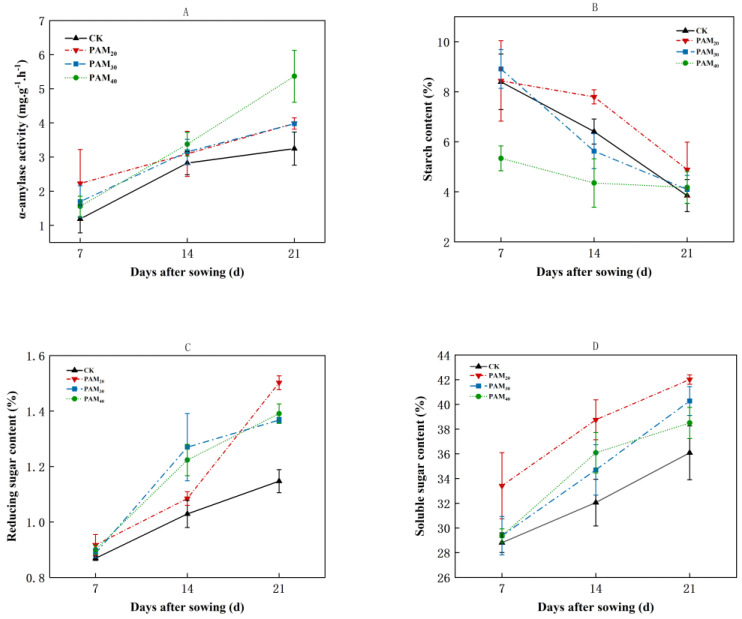
Effect of PAM treatment on α-amylase activity and starch, reducing sugar content, and soluble sugar content of sprouting seed potato. (**A**) α-amylase activity. (**B**) Starch content. (**C**) reducing sugar content. (**D**) Soluble sugar content. Note: CK, control (no polyacrylamide treatment); PAM_20_, 20 mg·L^−1^ polyacrylamide; PAM_30_, 30 mg·L^−1^ polyacrylamide; PAM_40_, 40 mg·L^−1^ polyacrylamide. The data represent the mean ± SD of three biological replicates.

**Table 1 plants-13-02796-t001:** Effect of PAM on seedlings plant height, stem diameter, and biomass.

PAM	Plant Height (cm)	Stem Diameter (cm)	Shoot FW (g)	Shoot DW (g)	Root FW (g)	Root DW (g)
CK	21.52 ± 0.89 bc	7.55 ± 0.41 b	29.12 ± 5.28 d	3.72 ± 0.12 b	6.25 ± 0.91 b	0.81 ± 0.19 b
PAM_20_	24.82 ± 0.92 a	9.06 ± 0.35 a	46.67 ± 6.05 ab	5.15 ± 0.41 a	7.01 ± 0.63 b	0.97 ± 0.13 a
PAM_30_	23.27 ± 0.99 ab	8.45 ± 0.31 ab	55.36 ± 9.23 a	5.69 ± 0.83 a	7.61 ± 0.46 a	1.05 ± 0.19 a
PAM_40_	21.83 ± 1.22 c	8.06 ± 0.42 ab	35.35 ± 2.82 c	4.12 ± 0.48 b	7.37 ± 0.89 a	0.99 ± 0.17 a

Note: CK, control (no polyacrylamide treatment); PAM_20_, 20 mg·L^−1^ polyacrylamide; PAM_30_, 30 mg·L^−1^ polyacrylamide; PAM_40_, 40 mg·L^−1^ polyacrylamide. Lowercase letter indicates a significant difference (*p* < 0.05) among treatments. The data represent the mean ± SD of three biological replicates.

## Data Availability

All data generated or analyzed during this study are included in this article. All data are available upon reasonable request.

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
