# Peer review of "Polyacrylamide Regulated Phytohormone Balance and Starch Degradation to Promote Seed-Potato Sprouting and Emergence"

_plants, 2024, doi:10.3390/plants13192796_

Round 1

Reviewer 1 Report

Comments and Suggestions for Authors

This work investigates the effect of different concentrations of polyacrylamide on seed-potato sprouting and emergence. The results are very interesting and well described. Some minor revisions should be done for this MS.

1. The main issue is insufficiently described methods especially ELISA assay and analysis of starch content. The methods should be described in detail.

2. The results on PAM-enhanced potato tuber respiration intensity should be discussed in Subsection 3.3 of the Discussion.

3. It would be better to mark the SubFigures in Figures 4, 9 and 10 with letter symbols (Figure 4 – A, B; Figures 9 and 10 – A, B, C, and D).

4. The Figures 9 and 10 would look better if the SubFigures A and B were placed next to each other and below them SubFigures C and D were placed next to each other rather than all the SubFigures were one below each other.

Comments on the Quality of English Language

Overall, the language is fine. Some minor revisions should be done.

For instance, Conclusion:

"Polyacrylamide treatment increases the content of growth-promoting phytohormones (IAA, CTK, and GA), reduces ABA content to break dormancy, and initiates sprouting."

The sentence is not grammatically correct.

Author Response

Dear Reviewer:

Thank you very much for your comments, which make me to renew revise our manuscript to improve its methods, figures discussion and discourse in order to more reasonable and the conclusion clear. We made a lot of modifications according to the comments, marked them in red font in the revised paper.

Comments 1. The main issue is insufficiently described methods especially ELISA assay and analysis of starch content. The methods should be described in detail.

Response1: We have added the ELISA assay to lines 391-395 of the paper and the starch content assay to lines 401-418 in the paper.

4.5 Quantitative analysis of phytohormones

Phytohormone levels were analyzed at four distinct stages of potato sprouting. Before treatment (BT), at germination initiation (GI), during germination (GS), and during vigorous growth (VG), which were 0, 20, 30,40 days after PAM treatment, respectively. The meristematic tissue (0.5 g) of the top bud and tuber bud eye tissue were frozen with liquid nitrogen and were ground in 4.5 mL 0.01 mol· L-1 phosphate buffer solution (pH 7.5). Then, the mixture was centrifuged (4 °C) at 12000 g for 10 min. The supernatant was used to quantify endogenous phytohormone levels using an enzyme-linked immunosorbent assay (Shanghai Enzyme Union Biotechnology Co., Ltd.).

4.6 Starch metabolism analysis

Sprouting tuber tissues were used to analyze starch metabolism. The 3,5-dinitrosalicylic acid colorimetric method was used to determine the activity of α-amylase and the reducing sugar content. Starch and soluble sugar contents were quantified using the anthrone-sulfuric acid colorimetric method.

Dried tuber tissues were ground and passed through a 0.15-mm sieve. The powdered sample (0.2 g) was placed in 6 mL 80% (v/v) ethanol and then incubated in 80°C water bath for 30 min. Then, the mixture was centrifuged (4 °C) at 4000 g for 5 min, the supernatant was collected, 6 mL of 80% ethanol was added to the residue, and the extraction was repeated twice. The three supernatants were collected and diluted to 25 mL using 80% ethanol. The resulting extract was used to determine the soluble and reducing sugar content by spectrophotometry at OD620 and OD540.

The ethanol-insoluble residue was then stored for starch extraction. The starch in the residue was released by boiling with 2 mL of distilled water for 15 min and cooling to 10°C. Then, the starch was hydrolyzed with 9.2 mol·L−1 HClO4 (2 mL) for 15 min. Four milliliter distilled water was added to the samples, which were centrifuged at 4000 g and 4°C for 10 min. The residue was extracted again using 4.6 mol·L−1 HClO4 (2 mL). The supernatants were combined and diluted with distilled water to 25 mL to obtain starch extract. The starch content was measured using a spectrophotometer at OD620.

Potato tuber tissues (1.0 g) ground using a chilled mortar containing 1 mL pre-cooled distilled water. The slurry was centrifuged at 3000 g at 4°C for 10 min. one milliliter supernatant was used to assay the α-amylase activity by 3,5-dinitrosalicylic acid colorimetric spectrophotometry at OD540.

Comments 2. The results on PAM-enhanced potato tuber respiration intensity should be discussed in Subsection 3.3 of the Discussion.

Response2: Subsection 3.4 of the Discussion were added to discuss the results on PAM-enhanced potato tuber respiration intensity. Line 337-355.

Comments 3. It would be better to mark the SubFigures in Figures 4, 9 and 10 with letter symbols (Figure 4 – A, B; Figures 9 and 10 – A, B, C, and D).

Response3: We've modified the graph mark in the article. For example, figure 4.

Comments 4. The Figures 9 and 10 would look better if the SubFigures A and B were placed next to each other and below them SubFigures C and D were placed next to each other rather than all the SubFigures were one below each other.

Response4: The Figures 9 and 10 have been rearranged according to your reviews. For example, figure 9.

Comments 5. Comments on the Quality of English Language Overall, the language is fine. Some minor revisions should be done. For instance, Conclusion:

"Polyacrylamide treatment increases the content of growth-promoting phytohormones (IAA, CTK, and GA), reduces ABA content to break dormancy, and initiates sprouting."

The sentence is not grammatically correct.

Response5: Thanks to your suggestions, we have modified it in the article (line426): Polyacrylamide treatment of potato tubers increases the content of growth-promoting phytohormones (IAA, CTK, and GA), reduces ABA content to break dormancy, and initiates sprouting.

Thank you and best regards.

Yours sincerely,

YinYuan Wen

2024.09.27

Reviewer 2 Report

Comments and Suggestions for Authors

The manuscript under consideration is devoted to the assessment of the effect of potato tuber treatments aimed at increasing their growth characteristics. In general, the problem under consideration is relevant and the results of the manuscript can be recommended for publication in the journal. However, there are a number of comments on this manuscript:

Comments:

1. Line 28 Solanum tuberosum should be written Solanum tuberosum L.

2. Line 64 The abbreviation IAA must be expanded

3. Comments on the figures:

Figure 1. In the figure caption, it should be written what a, b, c above the diagram columns mean, and it should also be written The data represent the mean ± SD of three biological replicates. ** p < 0.01.

Figure 2-10. The same comment for all figures: The caption of each figure should include a description of the abbreviations given in the diagrams.

CK, control (no polyacrylamide treatment), PAM20, 20 mg L-1 polyacrylamide; PAM30, 30 mg L-1 polyacrylamide; PAM40, 40 mg L-1 polyacrylamide. In the figure caption, it is necessary to write what a, b, c above the diagram columns mean, and also write The data represent the mean ± SD of three biological replicates. ** p < 0.01.

In Figures 4, 9, 10, the captions should describe in more detail what is presented in the figure A, B, etc. Letter designations, for example, Fig.10A,B are present in the text of the manuscript, but are absent in the figures themselves.

It is necessary to introduce uniformity in the figures, since in a number of figures the concentrations of polyacrylamide are designated as PAM20, PAM30, PAM40, and in Fig. 3 and 5 as 20,30,40.

In the caption to Table 1, it is necessary to add: The data represent the mean ± SD of three biological replicates. ** p < 0.01. In the figure caption, write what a, b mean.

Add the explanation control (no polyacrylamide treatment), PAM20, 20 mg L-1 polyacrylamide; PAM30, 30 mg L-1 polyacrylamide; PAM40, 40 mg L-1 polyacrylamide

4. Line 160-162 Auxin and CTK contents increased significantly with PAM20 and PAM30 at the GI stage, and the maximum contents obtained with PAM30 treatment were increased by 12.94 % and 47.04 %, respectively, compared to CK.

This sentence does not describe the results correctly. According to the graph, a significant increase with PAM20 and PAM30 treatment at the GI stage is observed only for CTK. There is no such pattern for auxins.

5. Line 192-193 The starch content in the PAM30 and PAM40 treatments were reduced by 12.2% and 32.1%, respectively, compared to CK.

It is not clear in this sentence on which day these changes occur. An amendment is required.

6. Line 200-202 The soluble sugar content significantly increased after PAM treatment, and the effect of PAM20 was the most significant, increasing by 15.98%, 20.90%, and 16.43% at the 7th, 14th and 21st days, respectively.

Judging by the graph, the figure of 16.43% on the 21st day is incorrect; the difference is significantly greater.

7. Line 214 The abbreviation CS2 must be deciphered.

8. Line 331 4.5 Quantitative analysis of phytohormones the name of the company enzyme-linked immunosorbent assay should be indicated.

9. Question on the methodology. By what criteria were the time of potato treatment with polyacrylamide and the concentration of the active substance selected? Were preliminary experiments conducted to select the optimal conditions for the experiment?

10. In conclusion, in the first sentence it should be added that the treatment was potatoes.

Polyacrylamide treatment of potato tubers increases the content of growth-promoting phytohormones (IAA, CTK, and GA), reduces ABA content to break dormancy, and initiates sprouting.

Author Response

Dear Reviewer:

Thank you very much for your comments, which make me to renew revise our manuscript to improve its structure and discourse in order to more reasonable and the conclusion clear. We made a lot of modifications according to the comments, marked them in red font in the revised paper.

Comments1. Line 28 Solanum tuberosum should be written Solanum tuberosum L.

Response1: Solanum tuberosum has been revised and be written Solanum tuberosum L.  (line28).

Comments 2. Line 64 The abbreviation IAA must be expanded.

Response2: The abbreviation IAA must be expanded: (line 65)

The ABA content in tubers is usually high at the onset of dormancy, peaks during deep dormancy, and declines during post-harvest storage, with a significant rise in Auxin(IAA)levels, which coincides with the break of dormancy.

Comments 3. Comments on the figures:

Comments 3.1 Figure 1. In the figure caption, it should be written what a, b, c above the diagram columns mean, and it should also be written The data represent the mean ± SD of three biological replicates. ** p < 0.01.

Response3.1:  Figure 1. Line 118-121.

Note:CK, control (no polyacrylamide treatment); PAM20, 20 mg·L-1 polyacrylamide; PAM30, 30 mg·L-1 polyacrylamide; PAM40, 40 mg·L-1 polyacrylamide. Lowercase letter above the diagram columns indicates a significant difference (p < 0.05) among treatments. The data represent the mean ± SD of three biological replicates.

Comments 3.2  Figure 2-10. The same comment for all figures: The caption of each figure should include a description of the abbreviations given in the diagrams.

CK, control (no polyacrylamide treatment), PAM20, 20 mg L-1 polyacrylamide; PAM30, 30 mg L-1 polyacrylamide; PAM40, 40 mg L-1 polyacrylamide. In the figure caption, it is necessary to write what a, b, c above the diagram columns mean, and also write The data represent the mean ± SD of three biological replicates. ** p < 0.01.

Response3.2:

Figure 2, 6,7:

Note:CK, control (no polyacrylamide treatment); PAM20, 20 mg·L-1 polyacrylamide; PAM30, 30 mg·L-1 polyacrylamide; PAM40, 40 mg·L-1 polyacrylamide.

Figure 3-5, 8-10:

Note:CK, control (no polyacrylamide treatment); PAM20, 20 mg·L-1 polyacrylamide; PAM30, 30 mg·L-1 polyacrylamide; PAM40, 40 mg·L-1 polyacrylamide. Lowercase letter above the diagram columns indicates a significant difference (p < 0.05) among treatments. The data represent the mean ± SD of three biological replicates.

Comments 3.3  In Figures 4, 9, 10, the captions should describe in more detail what is presented in the figure A, B, etc. Letter designations, for example, Fig.10 A, B are present in the text of the manuscript, but are absent in the figures themselves.

Response3.3:

Figures 4,the captions: Effect of PAM treatment on the sprouts length and diameter of potato sprouting in air. (A) Sprouts length. (B) Sprouts diameter.

Figures 9,the captions: Effect of PAM treatment on endogenous phytohormone content in potato tuber terminal buds. (A) Auxin content. (B) Gibberellin content. (C) Cytokinin content. (D) Abscisic acid content.

Figures 10,the captions: Effect of PAM treatment on α-amylase activity and starch, reducing sugar content, and soluble sugar content of sprouting seed potato. (A) É‘-amylase activity. (B) Starch content. (C) Reducing sugar content. (D) Soluble sugar content.

Comments 3.4  It is necessary to introduce uniformity in the figures, since in a number of figures the concentrations of polyacrylamide are designated as PAM20, PAM30, PAM40, and in Fig. 3 and 5 as 20,30,40.

Response3.4: We have provided a unified description of the PAM concentrations in all the figures and tables, which were designated as PAM20, PAM30, PAM40.

Comments 3.5  In the caption to Table 1, it is necessary to add: The data represent the mean ± SD of three biological replicates. ** p < 0.01. In the figure caption, write what a, b mean. Add the explanation control (no polyacrylamide treatment), PAM20, 20 mg L-1 polyacrylamide; PAM30, 30 mg L-1 polyacrylamide; PAM40, 40 mg L-1 polyacrylamide

Response3.5:  We have revised the caption of Table1 and Figures as the following in Table 1.

Note: CK, control (no polyacrylamide treatment); PAM20, 20 mg·L-1 polyacrylamide; PAM30, 30 mg·L-1 polyacrylamide; PAM40, 40 mg·L-1 polyacrylamide. Lowercase letter above the diagram columns indicates a significant difference (p < 0.05) among treatments. The data represent the mean ± SD of three biological replicates.

Comments 4. Line 160-162 Auxin and CTK contents increased significantly with PAM20 and PAM30 at the GI stage, and the maximum contents obtained with PAM30 treatment were increased by 12.94 % and 47.04 %, respectively, compared to CK.

This sentence does not describe the results correctly. According to the graph, a significant increase with PAM20 and PAM30 treatment at the GI stage is observed only for CTK. There is no such pattern for auxins.

Response4: Thanks for the suggestion. We conducted a statistical analysis on the experimental data again, and the results showed that, Auxin contents increased significantly with PAM20 and PAM30 at the GI stage, and the maximum contents obtained with PAM30 treatment were increased by 12.94 % compared to CK.

The data of auxin content recorded in the experiment is shown in the Table below.

The content of auxin in the GI period

PAM treatment

Auxin content(ug/g)

Average

5% significant level

CK

0.5618

0.5778

0.5662

0.5686

c

PAM20

0.6132

0.5985

0.5859

0.5992

b

PAM30

0.6437

0.6376

0.6453

0.6422

a

PAM40

0.5985

0.5778

0.5895

0.5886

b

Duncan Multiple Comparisons

PAM treatment

Average

PAM30

PAM20

PAM40

CK

PAM30

0.6422

0.0006

0.0002

0.0000

PAM20

0.5992

0.0430

0.2192

0.0060

PAM40

0.5886

0.0536

0.0106

0.0361

CK

0.5686

0.0736

0.0306

0.0200

Note:The lower triangle is the mean difference, and the upper triangle is the significant level

Comments 5. Line 192-193 The starch content in the PAM30 and PAM40 treatments were reduced by 12.2% and 32.1%, respectively, compared to CK.

It is not clear in this sentence on which day these changes occur. An amendment is required.

Response5: Thank you for your suggestions,the text has been revised (line220).

By the 14th day,the starch content in the PAM30 and PAM40 treatments were reduced by 12.2 % and 32.1 %, respectively, compared to CK.

Comments 6. Line 200-202 The soluble sugar content significantly increased after PAM treatment, and the effect of PAM20 was the most significant, increasing by 15.98%, 20.90%, and 16.43% at the 7th, 14th and 21st days, respectively.

Judging by the graph, the figure of 16.43% on the 21st day is incorrect; the difference is significantly greater.

Response6: Thanks for your suggestion. We conducted a statistical analysis on the experimental data again, and the results showed that, PAM20 increased the soluble sugar content by 16.43% at the 21st days.

The value (16.43%) is calculated using the formula: Increased percentage(%)=(A-B)/B*100%. A: PAM20 soluble sugar content at the 21st day, B: CK soluble sugar content at the 21st day.

The data of soluble sugar content recorded in the experiment is shown in the Table below.

Effect of PAM treatment on soluble sugar content of sprouting seed potato.

Treatment Days/d

PAM treatment

Soluble sugar content/%

Average/%

Increase the percentage/%

7

CK

29.4114

26.6257

30.3971

28.8114

15.98

PAM20

31.1971

28.2686

40.7829

33.4162

14

CK

30.8114

28.1829

37.1686

32.0543

20.90

PAM20

34.2257

41.5257

40.5114

38.7543

21

CK

42.2543

32.7829

33.2114

36.0829

16.43

PAM20

42.2543

42.7829

40.9971

42.0114

Comments 7. Line 214 The abbreviation CS2 must be deciphered.

Response7: The full name of the abbreviation CS2 has been added in paper (line246). Carbon disulfide (CS2)

Comments 8. Line 331 4.5 Quantitative analysis of phytohormones the name of the company enzyme-linked immunosorbent assay should be indicated.

Response8:  The company name of the enzyme-linked immunosorbent assay has been added in methods of quantitative analysis of phytohormones (line391-395).

Comments 9. Question on the methodology. By what criteria were the time of potato treatment with and the concentration of the active substance selected? Were preliminary experiments conducted to select the optimal conditions for the experiment?

Response9:  In this experiment, potato treatment with polyacrylamide is the seed potato dressing or coating by mixing 200 g potato mini-tubers with 300 ml PAM solution, and stirred for 5 min. Then the potatoes treated (including the PAM) were transferred to dark room for air-dried and used for sprouting investigations. So, we think that the time (5 min) of potato treatment with polyacrylamide is sufficient.

About the concentration selection of the polyacrylamide based on our previous research in foxtail millet (Ke, Z.J., Yin, M.Q., Wen, Y.Y., et al. Effects of polyacrylamide seed soaking on seed germination and drought resistance of millet seedlings under drought stress. J. Nucl. Agric. Sci. 2015,29,563-570.) and the preliminary experiments in potatoes. The PAM concentration is too higher to stir evenly in the mixture of tubers and PAM when the concentration exceeds 40 mg·L-1. On the contrary, when the concentration of PAM is below 10 mg·L-1, it is difficult for PAM to dress or coat to the tubers. So, the concentration of 20, 30, and 40 g · L-1 PAM was used in this experiment.

Comments 10. In conclusion, in the first sentence it should be added that the treatment was potatoes.

Polyacrylamide treatment of potato tubers increases the content of growth-promoting phytohormones (IAA, CTK, and GA), reduces ABA content to break dormancy, and initiates sprouting.

Response10: Thanks for your suggestion. We have revised the sentence as your reviewers (line426).

 Polyacrylamide treatment of potato tubers increases the content of growth-promoting phytohormones (IAA, CTK, and GA), reduces ABA content to break dormancy, and initiates sprouting.

Thank you and best regards.

Yours sincerely,

YinYuan Wen

2024.9.27

Reviewer 3 Report

Comments and Suggestions for Authors

To the Authors

Review of the paper entitled “Polyacrylamide regulated phytohormone balance and starch 2 degradation to promote seed-potato sprouting and emergence”. The advantage of this manuscript is the search for technologies to accelerate and equalize potato tuber emergence, which is valuable in the conditions of growing very early and early potato varieties and to accelerate the emergence of starchy medium-early and late potato varieties in order to accumulate starch faster.

1.      The paper describes the practical problem of slow germination of potato tubers and its impact on production, and then proposes polyacrylamide as a potential solution. The introduction provides a good overview of the agricultural context, including the challenges related to controlling tuber emergence, and indicates the importance of promoting rapid and uniform potato germination to improve yields. The purpose of the paper is well explained.

2.      The methodology of the paper is well developed and applied.

3.      Interpretation of the results is in line with the presented data, paying attention to key indicators such as germination rate, seedling vigor, tube respiration, phytohormone levels and starch conversion.

4.      The discussion of the results is detailed and based on research results, which makes it solid and comprehensive. However, it could be more in-depth by:

-       References to recent literature. Citing relevant sources (e.g. on phytohormones or the effect of PAM on germination) strengthens the argument. Ensure that recent scientific research is included, especially in areas such as biotechnology or nanotechnology.

-       Long-term effects of PAM use in agricultural practice, the study of which could provide a more complete knowledge of the potential benefits and risks.

-       Practical perspective: The discussion suggests that PAM can be effective as a material stimulating potato germination. However, it would be good to extend the analysis to include practical aspects of its implementation in agriculture. Are there economic or environmental barriers that could limit the wider use of PAM in commercial potato production?

-       Recommendations for further research: The authors suggest the need for further research on the mechanisms of phytohormone coordination. This is a good direction, but it would be worthwhile to formulate more precise research hypotheses or suggestions for future studies, e.g. what hormonal or molecular mechanisms should be investigated to obtain a more complete picture of PAM action.

Discussion is well conducted, but it could be enriched with an analysis of the limitations of the study, as well as more practical implications of the results and recommendations for future research.

5.      The conclusions are correct, they summarize the main findings of the study well and clearly and are consistent with the results presented earlier. However, they could be extended to include potential challenges related to the implementation of this solution on an industrial scale.

Detailed comments:

6.      Line 242, 249, 295 – please do not use the term “correlation” – because the authors of this manuscript did not present a correlation analysis in the paper.

7.      Line 305 – please complete the description of the Jinshu 16 potato variety, to which earliness group does it belong, the edible or industrial form?

8.      Line 325 – is the planting depth of minitubers weighing 15-25 g to a depth of 10 cm appropriate?

9.      Line 349. The application should be reworded and based on the obtained research results.

10.   In lines 350-355 the authors describe the effect of Polyacrylamide on potato minitubers. I propose moving the effect of Polyacrylamide on the physiology of potato emergence to the "Introduction" chapter.

11.   The authors use a large number of abbreviations. They should be placed at the end of the manuscript.

After the above corrections have been made, in my opinion this manuscript is suitable for publication in MDPI Plants.

Author Response

Dear Reviewer:

Thank you very much for your comments, which make me to renew revise our manuscript to improve its structure and discourse in order to more reasonable and the conclusion clear. We made a lot of modifications according to the comments, marked them in red font in the revised paper.

Comments 1. The paper describes the practical problem of slow germination of potato tubers and its impact on production, and then proposes polyacrylamide as a potential solution. The introduction provides a good overview of the agricultural context, including the challenges related to controlling tuber emergence, and indicates the importance of promoting rapid and uniform potato germination to improve yields. The purpose of the paper is well explained.

Response1: Thanks for your review.

Comments 2. The methodology of the paper is well developed and applied.

Response2: Thanks for your review.

Comments 3. Interpretation of the results is in line with the presented data, paying attention to key indicators such as germination rate, seedling vigor, tube respiration, phytohormone levels and starch conversion.

Response3: Thanks for your review.

Comments 4. The discussion of the results is detailed and based on research results, which makes it solid and comprehensive. However, it could be more in-depth by:

Comments 4.1 - References to recent literature. Citing relevant sources (e.g. on phytohormones or the effect of PAM on germination) strengthens the argument. Ensure that recent scientific research is included, especially in areas such as biotechnology or nanotechnology.

Response4.1: We add the content about nanotechnology at Line 250-251.

 Nano-priming is a considerably more effective technology that helps to improve seed germination, seedlings growth, and yield in agriculture [27].

Comments 4.2-Long-term effects of PAM use in agricultural practice, the study of which could provide a more complete knowledge of the potential benefits and risks.

Response4.2: We add the content about the knowledge of the potential benefits and risks at Line 266-269.

Additionally, PAM is likely to be categorized as a low-risk chemical when employed as seed priming agent or soil amendment, while further experiments are crucial for the po-tential benefits and risks assessments [17].

Comments 4.3-Practical perspective: The discussion suggests that PAM can be effective as a material stimulating potato germination. However, it would be good to extend the analysis to include practical aspects of its implementation in agriculture. Are there economic or environmental barriers that could limit the wider use of PAM in commercial potato production?

Response4.3: Thanks for your suggestion. These issues are important and more attention should be paid to it. We will analysis the PAM’s practical aspects, and the economic or environmental barriers were discussed. At this manuscript, the characteristics and application potential of PAM were described in Introduction (Line 75-81). The potential risks were supplemented in discussion 3.1(Line 266-269).

Comments 4.4-Recommendations for further research: The authors suggest the need for further research on the mechanisms of phytohormone coordination. This is a good direction, but it would be worthwhile to formulate more precise research hypotheses or suggestions for future studies, e.g. what hormonal or molecular mechanisms should be investigated to obtain a more complete picture of PAM action.

Response4.4: This is an excellent recommendation. We will further investigate the molecular mechanisms of hormones and their interactions in PAM promoting potato sprouting. Thanks for your great ideas.

Comments 4.5-Discussion is well conducted, but it could be enriched with an analysis of the limitations of the study, as well as more practical implications of the results and recommendations for future research.

Response 4.5:  We add the discussion at Line 354-355 (about hormones interactions) and Line 316-318 (about respiration).

Comments 5. The conclusions are correct, they summarize the main findings of the study well and clearly and are consistent with the results presented earlier. However, they could be extended to include potential challenges related to the implementation of this solution on an industrial scale.

Response5: The potential challenges were added in conclusion, Line 434-437.

The potential challenges to industrial-scale application of PAM-promoting sprouting in potatoes were carefully considered, including (i) differences in response among varieties, (ii) the compatibility of PAM with seed coating agents and plant regulators.

Detailed comments:

Comments 6. Line 242, 249, 295 – please do not use the term “correlation” – because the authors of this manuscript did not present a correlation analysis in the paper.

Response 6: The three term correlation in Line 242, 249, 265 were all described the References. We have modified these sentences.

Line 242, The ABA content of different potato varieties showed a continuous decline during storage, but the absolute ABA levels did not affect the potato sprouting [29]. Line 279

Line 249, Higher levels of GA significantly promote vigorous sprout growth. Line 285

Line 265, The breaking dormancy of potato tubers coincides with a significant rise in free IAA in the eyes [33]. Line 301

Comments 7. Line 305 – please complete the description of the Jinshu 16 potato variety, to which earliness group does it belong, the edible or industrial form?

Response7: We modified the description of Jinshu 16 potato variety (line358-360).

Jinshu 16 is a medium-late maturing potato variety. its dry matter and starch content were 22.3% and 15.6%, respectively, which were used as processing potatoes and fresh vegetable potatoes.

Comments 8. Line 325 – is the planting depth of minitubers weighing 15-25 g to a depth of 10 cm appropriate?

Response 8:  The planting depth of minitubers weighing 15-25 g to a depth of 10 cm, the reasons were to simulate potato field cultivate. Seconds, potato seedlings can emergence normally at 10 cm planting depth in our experiment. So, we think it’s appropriate.

Comments 9. Line 349. The application should be reworded and based on the obtained research results.

Response9: Thanks for your review. We reworded the application based on the obtained research results.

Line 433-434: This study provides a promising approach for promoting potato tuber sprouting and seedling growth, with broad practical application in promoting potato tuber sprouting.

Comments 10. In lines 350-355 the authors describe the effect of Polyacrylamide on potato minitubers. I propose moving the effect of Polyacrylamide on the physiology of potato emergence to the "Introduction" chapter.

Response10: We add the “seedlings emergence” to the introduction in Line 92.

In this study, different concentrations of PAM were used as seed potato dressings, and their effects on sprouting, seedling emergence and growth, and biomass were investigated.

Comments 11. The authors use a large number of abbreviations. They should be placed at the end of the manuscript.

Response11: The abbreviations were added at the end of the manuscript. (Line 455-468)

Abbreviations

ABA: Abscisic acid

BT: Before treatment

CK: Control (no polyacrylamide treatment)

CTK: Cytokinin

CS2: Carbon disulfide

DW: Dry weight

FW: Fresh weight

GA: Gibberellin

GI: Germination initiation

GS: Germination stage

IAA: Auxin

PAM: Polyacrylamide

VG: Vigorous growth

Round 2

Reviewer 2 Report

Comments and Suggestions for Authors

All comments have been taken into account by the authors. The manuscript can be recommended for publishing.